# A Luminescence-Based Human TRPV1 Assay System for Quantifying Pungency in Spicy Foods

**DOI:** 10.3390/foods10010151

**Published:** 2021-01-13

**Authors:** Minami Matsuyama, Yuko Terada, Toyomi Yamazaki-Ito, Keisuke Ito

**Affiliations:** Department of Food and Nutritional Sciences, Graduate School of Integrated Pharmaceutical and Nutritional Sciences, University of Shizuoka, 52-1 Yada, Suruga-ku, Shizuoka 422-8526, Japan; s19215@u-shizuoka-ken.ac.jp (M.M.); yukoterada@u-shizuoka-ken.ac.jp (Y.T.); imyot1207@gmail.com (T.Y.-I.)

**Keywords:** pungency evaluation, TRPV1, luminescence assay, spicy foods, flavor evaluation

## Abstract

The quantitation of pungency is difficult to achieve using sensory tests because of persistence, accumulation, and desensitization to the perception of pungency. Transient receptor vanilloid 1 (TRPV1), which is a chemosensory receptor, plays a pivotal role in the perception of many pungent compounds, suggesting that the activity of this receptor might be useful as an index for pungency evaluation. Although Ca^2+^-sensitive fluorescence dyes are commonly used for measuring human TRPV1 (hTRPV1) activity, their application is limited, as foods often contain fluorescent substances that interfere with the fluorescent signals. This study aims to design a new pungency evaluation system using hTRPV1. Instead of employing a fluorescent probe as the Ca^2+^ indicator, this assay system uses the luminescent protein aequorin. The luminescence assay successfully evaluated the hTRPV1 activity in foods without purification, even for those containing fluorescent substances. The hTRPV1 activity in food samples correlated strongly with the pungency intensity obtained by the human sensory test. This luminescence-based hTRPV1 assay system will be a powerful tool for objectively quantifying the pungency of spicy foods in both laboratory and industrial settings.

## 1. Introduction

Pungency, which is an oral sensation, is characterized by persistence, accumulation, and desensitization [1]. Pungency is one of the most important determinants of the taste of food. However, a quantitative evaluation of pungency is difficult to achieve using standard sensory tests. The Scoville unit is an index employed for evaluating the pungency of chili peppers. The unit indicates the number of times a sample has to be diluted such that the pungency is not perceived [2]. Although the Scoville unit is the most classic measure of pungency and is still in use today, it has drawbacks, including a poor accuracy, subjectivity, a low reproducibility, and taste fatigue in subjects. Therefore, a new pungency evaluation method is needed to enable a more detailed analysis of food pungency.

Transient receptor potential (TRP) channels are a superfamily of cation trans-membrane proteins that are expressed in many tissues and respond to many sensory stimuli [3,4]. TRP channels play a role in sensory signaling for taste, thermo-sensation, mechanosensation, and nociception [3]. TRP vanilloid 1 (TRPV1), TRP ankyrin 1 (TRPA1), and TRP melastatin 8 (TRPM8) are important for the flavor perception of spices and herbs [5]. Typically, TRPV1 is activated by pungent compounds such as capsaicinoids (in chili pepper), piperines (in pepper), shogaols and gingerols (in ginger), and sanshools (in Japanese pepper) [5,6,7,8,9]. The activation of TRPV1 results in the stimulation of sensory nerve endings in the oral cavity to elicit a burning sensation [10]. Human TRPV1 (hTRPV1) has been shown to play a pivotal role in the perception of pungency in foods [3].

Ca^2+^-sensitive fluorescence dyes such as Fluo-4 and Fluo-8 can be used to measure hTRPV1 activity because hTRPV1 activation induces an increase in the intracellular Ca^2+^ concentration [11,12,13]. However, Ca^2+^-sensitive fluorescence indicators are not easily applied to food-derived samples because foods often contain fluorescent substances, such as Maillard reaction products, polyphenols, vitamins, and chlorophyll [14,15], which can interfere with the detection of fluorescent probe signals. In such cases, hTRPV1 activity would have to be assessed after the purification of each pungent compound. However, it would be difficult to assess the hTRPV1 activity of a food that can be compositely formed by multiple pungent substances. Indeed, synergistic effects between TRPV1 agonists on TRPV1 activation have been reported [16,17]. These reports indicate that the total TRPV1 activity of a food is not necessarily the sum of the activities of the individual TRPV1 agonists contained therein.

To design a new pungency evaluation system that is applicable to spicy foods with auto-fluorescence, this study aims to develop an advanced hTRPV1 assay system. We investigate the use of the luminescent protein aequorin [18]. Active aequorin is composed of apoaequorin and coelenterazine in the presence of molecular oxygen [19]. The binding of Ca^2+^ to apoaequorin induces a conformational change in the protein that results in the oxidation of coelenterazine to form coelenteramide and the emission of blue light. Therefore, the light emission can be used to assess changes in the Ca^2+^ concentration. Photoproteins have been used as Ca^2+^ indicators in assay systems for G-protein-coupled receptors (GPCRs) and ion channels [20,21,22].

## 2. Experimental Section

### 2.1. Materials

Capsaicin was purchased from FUJIFILM Wako Pure Chemical Industries (Osaka, Japan). Coelenterazine *hcp*, *ip*, *f*, *cp*, *n*, *h*, *i*, and *fcp* were obtained from Biotium, Inc. (Fremont, CA, USA). T-REx293 cells were purchased from Invitrogen (Carlsbad, CA, USA). The pHEK293 Ultra Expression Vector I was purchased from Takara Bio Inc. (Shiga, Japan). An artificial gene encoding aequorin protein with codon usage optimized for expression in human cells was chemically synthesized by Eurofins Genomics (Tokyo, Japan). The nucleotide sequence of aequorine is shown in Appendix A. The aequorin gene was inserted into the multiple cloning site of pHEK293 Ultra Expression Vector I.

### 2.2. Analysis of the Fluorescence Spectrum of Foods

The fluorescence spectrum of 15 foods and six types of spicy instant noodles was examined. Miso, curry, hot pepper paste, chocolate, Worcestershire sauce, oyster sauce, hot chocolate, coffee, soy sauce, maple syrup, tea, vegetable juice, red wine, soy milk, and dashi were used as the food samples. These 15 foods were purchased from a local supermarket. The spicy instant noodles were purchased from Myojo Foods Co., Ltd. (Tokyo, Japan), Sugakiya Foods Co., Ltd. (Aichi, Japan), Toyo Suisan Kaisha, Ltd. (Tokyo, Japan), and Nongshim Japan Co., Ltd. (Tokyo, Japan). The spicy noodles were cooked according to the instructions and the filtered broth was used as the sample. Curry and chocolate were diluted with dimethyl sulfoxide (DMSO) to 10% (*w*/*v*), as they have a high oil and fat content. The other samples were diluted with distilled water to 10% (*w*/*v*). Samples with turbidity or precipitate were filtered using an Ultrafree-MC Centrifugal Filter (pore size 0.22 µm, Merck Millipore, Burlington, MA, USA). All samples were dispensed at 200 µL/well into 96-well plates. The fluorescence spectra under the wavelength used for the Fluo-8 fluorescent Ca^2+^ probe (excitation wavelength, 490 nm) were measured using a FlexStation III microplate reader (Molecular Devices, Inc., San Jose, CA, USA) at 37 °C. The analysis software SoftMaxPro 7 equipped with FlexStation III was used for fluorescence spectrum analysis.

### 2.3. Culture of Cells Expressing hTRPV1

T-REx293 cells with a tetracycline-inducible gene expression system stably expressing hTRPV1 were used [23]. Details of the establishment of a stable cell line are described in our previous report [24]. The cells were cultured at 37 °C in the presence of 5% CO_2_ in Dulbecco’s modified Eagle’s medium high glucose (Sigma-Aldrich, St. Louis, MO, USA) containing 10% fetal bovine serum (Ireland origin; Biowest, Nuaillé, France), and 1% antibiotic-antimycotic, blastcidin, and zeocin. The cells were passaged 2–3 times a week and used for up to 48 passages.

### 2.4. Fluorescence-Based hTRPV1 Assay

Details of the hTRPV1 assay method are described in our previous report [24]. Briefly, hTRPV1-expressing cells were seeded in 96-well tissue culture-treated plates (Corning, Corning, NY, USA) at 4.0 × 10^4^ cells/well. After the addition of 1 µg/mL tetracycline (Invitrogen) to induce hTRPV1 expression, the cells were incubated at 37 °C in the presence of 5% CO_2_ for 19–20 h. The medium was removed, and the cells were washed with the assay buffer. Fluo-8 AM (3 µM) (AAT Bioquest, Sunnyvale, CA, USA) diluted with assay buffer was gently added (50 µL/well) to the plate so as not to detach the cells and incubated at 37 °C in the presence of 5% CO_2_ for 1 h. After removing the Fluo-8 AM solution and washing with assay buffer (100 µL/well), assay buffer (180 µL/well) was added to the cells. Test samples were added to each well using FlexStation III, and the changes in the fluorescence intensity at 37 °C were measured (wavelengths: excitation, 490 nm; cutoff, 515 nm; emission, 525 nm). hTRPV1 activity was calculated using the equation response = ΔF/F_0_, where F_0_ (baseline) is defined as the mean fluorescence value at 0–30 s before sample addition, and ΔF is calculated by subtracting F (the signal intensity, the highest fluorescence value at 30–150 s after sample administration) from F_0_. T-REx293 cells not expressing the hTRPV1 gene were used as the parent strain. The half maximal (50%) effective concentration (EC_50_) values of each sample were estimated from the dose–response curves prepared using GraphPad PRISM ver. 4.03 (GraphPad Software, Inc., San Diego, CA, USA). The reproducibility was confirmed in 2–3 separate experiments, and data are presented as the mean ± standard error (SE).

The assay buffer was prepared by mixing 50 mL of Hank’s BSS, 10 mL of 1 M 2-[4-(2-hydroxyethyl)-1-piperazinyl]ethane-sulfonic acid (HEPES), 20 mL of 25 mM CaCl_2_, 5 mL of 10% albumin from bovine serum, and 410 mL of sterilized water. To prepare the assay buffer containing miso, miso was added to the assay buffer to concentrations of 5, 10, and 20% (10 times the final concentration) and filtered with a centrifugal filter (pore size, 0.22 µm; Merck Millipore). Capsaicin dissolved in DMSO was diluted to concentrations of 0.1 nM, 1 nM, 3 nM, 10 nM, 30 nM, 0.1 µM, 0.3 µM, 10 µM, and 100 µM (10 times the final concentration) using assay buffer or the assay buffer containing miso on the compound plate. The capsaicin solution was administered to the cells at final concentrations of 0.01 nM, 0.1 nM, 0.3 nM, 1 nM, 3 nM, 10 nM, 30 nM, 1 µM, and 10 µM.

### 2.5. Luminescence-Based Assay of hTRPV1 Activity

hTRPV1-expressing cells were seeded on 6-well plates at 3.0 × 10^5^ cells/well and incubated at 37 °C in the presence of 5% CO_2_ for 19–20 h. An aequorin expression plasmid (4 µg/well) was transfected into the cells using Lipofectamine 2000 (Invitrogen). After incubation at 37 °C in the presence of 5% CO_2_ for 5 h, the transfected cells were treated with trypsin and seeded into 96-well plates at the same density used in the fluorescence method (4.0 × 10^4^ cells/well). Following the addition of 1 µg/mL tetracycline to induce hTRPV1 expression, the cells were incubated at 37 °C in the presence of 5% CO_2_ for 19–20 h. After removal of the medium, the cells were washed with assay buffer (50 μL/well). Coelenterazine (10 μM in assay buffer) was gently added so that the cells did not detach, followed by incubation at 37 °C in the presence of 5% CO_2_ for 2 h. After removing the coelenterazine solution and washing with assay buffer (100 μL/well), assay buffer (180 μL/well) was added to the cells. The change in the luminescence intensity after sample administration was measured by FlexStation III (detection wavelength, 466 nm). SoftMaxPro 7 software was used to analyze the changes in light emission. The activation of hTRPV1 was determined using the equation response = ΔL/L_0_, where L_0_ (baseline) is defined as the mean luminescence at 0–20 s before sample addition, and ΔL is calculated by subtracting L (the signal intensity, the highest luminescence value at 20–60 s after sample administration) from L_0_. Each experiment was repeated at least two or three times to check the reproducibility, and the data are presented as the mean ± SE.

Assay buffer, assay buffer containing miso, and the capsaicin solution were prepared in the same manner as that employed for the fluorescence assay. Broths from the spicy instant noodles were filtered as shown in Section 2.2. The filtered samples were diluted with assay buffer to concentrations of 0.01, 0.1, 1, 3, 10, 30, and 70% on compound plates and added to the cells to final concentrations of 0.001, 0.01, 0.1, 0.3, 1, 3, 7, and 10%.

### 2.6. Sensory Evaluation of Spicy Instant Noodle Pungency

The pungency of six types of spicy instant noodles was determined by sensory evaluation by a panel of 12 adults (2 men and 10 women) aged 21–40 years. The sensory panelists were untrained consumers. Each panelist evaluated all six samples (*n* = 12). The outline of the research was explained to each panelist, and approval for data use was obtained from all panelists. In a preliminary study, the following experimental condition that did not show a carry over effect in each sample flavor was determined. The participants were not allowed to eat spicy food for 24 h before the test and were asked to refrain from eating, smoking, brushing teeth, taking painkillers, applying perfume, or talking during and at least 1 h before testing. Each sample was served at room temperature in a clear plastic cup (10 mL) labeled with a three-digit random code. Their broth samples were diluted to 10% with distilled water before use, as they were too spicy for allowing the panelists to evaluate several samples. The test was carried out in a sensory evaluation room (room temperature 25 °C, humidity 47%) under red light. Each panelist received samples in their own booth. After rinsing the mouth with distilled water at least three times, the subjects kept 0.4 ppm capsaicin solution (standard solution) in their mouth for 10 s. Ten seconds after spitting the sample out, the panelists noted their sense of pungency of capsaicin solution as the maximum pungency. During a 2.5 min break, the panelists rinsed their mouths with distilled water until flavor sensation disappeared. After washing the mouth three times with distilled water, the panelists took a sample of broth into their mouths and kept it there for 10 s. Ten seconds after spitting the sample out, the panelists rated their sense of pungency on a 100-mm visual analog scale (VAS) ranging from “no sense of pungency” at the left end to “sense of pungency of 0.4 ppm capsaicin solution” at the right end. The VAS was used as described in a previous study [25,26]. Each subject evaluated three samples/day for 2 days (total of six samples).

### 2.7. Statistical Analysis

An unpaired *t*-test was used for statistical analysis to examine whether Ca^2+^ responses in hTRPV1-expresssing cells significantly differed from those in parent cells not expressing hTRPV1. GraphPad PRISM ver 4.03 (GraphPad Software, Inc.) was used for the unpaired *t*-test. The dose–response curves were prepared using GraphPad PRISM ver 4.03 (GraphPad Software, Inc.).

## 3. Results and Discussion

### 3.1. Fluorescence Spectrum of Tested Foods

Ca^2+^ imaging using fluorescence dyes is one of the major methods employed for measuring the activation of TRP channels [27,28,29]. At the beginning of this study, we examined the fluorescence spectra of 15 foods (miso, curry, hot pepper paste, chocolate, Worcestershire sauce, oyster sauce, hot chocolate, coffee, soy sauce, maple syrup, tea, vegetable juice, red wine, soy milk, and dashi) under the wavelength used for the representative Ca^2+^-sensitive fluorescence probe Fluo-8 (excitation wavelength: 490 nm) (Figure 1). Ten of the foods elicited strong fluorescence at the wavelength used for the fluorescence probes (Ex, 490 nm; Em, 525 nm). Miso, curry, hot pepper paste, chocolate, and Worcestershire sauce showed particularly strong fluorescence. Miso, curry, hot pepper paste, Worcestershire sauce, and soy sauce are reported to contain Maillard reaction products [30,31,32,33,34,35]. As Maillard reaction products are known to exhibit fluorescence [14,15], melanoidin, which is a brown pigment, generated during the Maillard reaction is the likely source of the fluorescence in these foods. Fluorescence by polyphenols has also been reported [36]. Since chocolate and hot chocolate contain Maillard reaction products and polyphenols [33,37], these compounds are thought to be responsible for the observed fluorescence. Because multiple food components emit strong fluorescence, evaluating hTRPV1 activity using the fluorescence method in the presence of these components would be difficult, prompting our development of a luminescence-based assay system that does not use fluorescent probes.

### 3.2. Evaluation of hTRPV1 Activity Using Fluorescence and Luminescence-Based Assay Systems

In a typical fluorescence hTRPV1 assay, the activation of hTRPV1 expressed on the surface of T-REx293 cells induces Ca^2+^ influx into the cells (Figure 2a). The channel activity is detected using a Ca^2+^-sensitive fluorescent probe such as Fluo-8 (Figure 2b). A concentration–response curve of capsaicin for hTRPV1 obtained in the fluorescence assay is shown in Figure 2c. However, Figure 1 indicates that measuring hTRPV1 activity by the fluorescence method would be difficult in foods with auto-fluorescence. Therefore, in this study, we constructed a luminescence assay system using aequorin, which is a Ca^2+^-sensitive luminescence protein (Figure 2d). After the transfection of an aequorin-expression plasmid into hTRPV1-expressing cells, coelenterazine, which is a luminescent substrate, was loaded into the cells. The introduction of a TRPV1 agonist to the cells activates hTRPV1, inducing Ca^2+^ influx into the cells. Ca^2+^ binds to apoaequorin, which in turn catalyzes the oxidation of coelenterazine, producing luminescence. A representative chart and a dose–response curve of capsaicin for hTRPV1 are shown in Figure 2e,f, respectively. Both the fluorescence and luminescence assays yielded the concentration-dependent activation of hTRPV1 to capsaicin. The EC_50_ value of capsaicin for hTRPV1 was 36.7 nM in the luminescence assay. Although this value was slightly higher than that of the fluorescence assay (4.4 nM), the EC_50_ values were comparable to those of previous reports (EC_50_, 3–60 nM) [27,38]. The increase over the baseline (maximum response intensity) of the luminescence assay was 10 times higher than that of the fluorescence assay (Figure 2c,f). It was suggested that since the luminescence assay does not require excitation light, this assay has a lower background and higher sensitivity than the fluorescence assay. These results show that the luminescence assay using aequorin is applicable to the measurement of hTRPV1 activity and has a higher sensitivity than the fluorescence method using Fluo-8.

### 3.3. Comparison of the hTRPV1 Activity of Coelenterazines with Different Luminescence Characteristics

We then asked which type of coelenterazine, which is a substrate for aequorin, is most suitable for the luminescence assay. We compared the level of 10 µM capsaicin-induced hTRPV1 activity for eight types of coelenterazine (*hcp*, *ip*, *f*, *cp*, *n*, *h*, *i*, and *fcp*) (Figure 3a). Coelenterazine *hcp* and *cp* yielded the highest hTRPV1 response. The relationship between the intensity of the hTRPV1 response and four luminescence characteristics of coelenterazine (emission maximum, half-rise time, relative intensity, and relative luminescence capacity) [39,40] was analyzed (Figure 3b–e). The emission maximum, half-rise time, and relative intensity did not seem to be associated with the level of hTRPV1 activity (Figure 3b–d). In contrast, the relative luminescence capacity was suggested to be positively associated with the level of hTRPV1 activity: A higher relative luminescence capacity resulted in higher hTRPV1 activity (Figure 3e). The relative luminescence capacity indicates the yield of aequorin produced from apoaequorin [39]. A higher relative luminescence capacity results in a larger total light emission, explaining the correlation between the index and the level of hTRPV1 activity. Since native coelenterazine is reported to possess the highest relative luminescence capacity (1.0) of the known coelenterazines [39], it might have a higher sensitivity in the luminescence assay. An examination using native coelenterazine will be conducted in a future study. Since the response to capsaicin was greater for *hcp* and *cp* than the other coelenterazines, their use might provide a greater assay sensitivity. However, these reagents are expensive. Because we want to develop a luminescence assay system for food-derived samples that can be used routinely, we selected coelenterazine *h,* as it induced a sufficient hTRPV1 response to capsaicin, yet costs less than coelenterazine *hcp* or *cp*.

### 3.4. Evaluation of Capsaicin hTRPV1 Activity in the Presence of Fluorescent Substances

Various foods elicited strong fluorescence at the wavelength used in the fluorescence method (Figure 1). We investigated whether the fluorescence and luminescence assays can accurately detect hTRPV1 activity in samples containing fluorescent substances. Miso, which is a popular seasoning in Asia, was used as a model food because it emits strong fluorescence (Figure 1). hTRPV1 activation by capsaicin was measured using the fluorescence and luminescence method in the presence of 0, 0.5, 1, and 2% of miso, and the resulting concentration–response curves were compared. As higher concentrations of miso resulted in a higher background, the fluorescence assay could not assess dose-dependent hTRPV1 activation by capsaicin in the presence of ≥1% miso (Figure 4a). In contrast, the luminescence assay effectively demonstrated capsaicin concentration-dependent activation of hTRPV1, showing that this method was barely affected by 0.5–2% miso (Figure 4b). The result also showed that the miso did not activate hTRPV1. Therefore, the luminescence assay detects hTRPV1 activity, even in the presence of fluorescent substances. Miso soup contains about 10% miso. We observed that 1% miso disrupted the hTRPV1 fluorescence assay, representing a concentration one-tenth that of ordinary miso soup. Accordingly, the luminescence assay would be useful for assessing samples that contain a fluorescent ingredient such as miso.

### 3.5. Relationship between hTRPV1 Activity and the Sensory Evaluation of Spicy Instant Noodles

We then examined the following two points: (1) Whether the luminescence method is applicable for measuring hTRPV1 activation in spicy foods with auto-fluorescence, and (2) whether hTRPV1 activity correlates with the sensory evaluation of pungency. Six types of spicy instant noodles containing pungent ingredients, such as chili pepper, garlic, Chinese chive, leek, rajan, and chili bean sauce, were used as sample foods. Because all of the noodle soups elicited strong fluorescence at the wavelength used for the fluorescence assay (Figure 5a), the hTRPV1 activity could not be accurately measured using the fluorescence method. hTRPV1 activity was thus measured using the luminescence assay. The Ca^2+^ response of hTRPV1-expressing cells elicited by the noodle broths was significantly higher than that of parent cells not expressing hTRPV1, indicating that the response was attributable to hTRPV1 activity (Figure 5b). Moreover, all samples induced concentration-dependent hTRPV1 responses (Figure 5c). These results show that the luminescence assay is able to evaluate hTRPV1 activity in foods without purification, even if they contain fluorescent substances.

The relationship between hTRPV1 activity and the sensory evaluation of pungency was analyzed (Figure 5d,e). The hTRPV1 activity and pungency intensity were plotted on the x and y axis, respectively (Figure 5e). The regression curve shown in Figure 5e yielded R^2^ (decision coefficient) = 0.92, indicating a strong correlation between the hTRPV1 activity and pungency intensity at the concentrations tested. Collectively, these findings show that the luminescence-based assay using aequorin can be used to assess the hTRPV1 activity in foods with auto-fluorescence and that the hTRPV1 activity obtained by the assay system strongly correlates with the pungency intensity in the human sensory test.

One method employed for predicting the pungency of foods is quantifying the pungent compounds in those foods. There have been studies investigating whether quantifying a variety of pungent compounds enables the prediction of food pungency. For example, Dierkes et al. reported that the bitterness and pungency of olive oil correlated with the content of six taste compounds, including oleuropein aglycon, which is a TRPV1 agonist [41,42]. In contrast, Schneider et al. showed that the pungency intensity of salsa does not necessarily correlate with the content of capsaicin and dihydrocapsaicin, which are TRPV1 agonists [43], even though these compounds are reported as the main pungent components of salsa [44]. These reports suggested that quantifying individual pungent substances does not necessarily predict the pungency intensity of foods.

Food ingredients were shown in each noodle sample (A–F), and spice-related components were as follows: (A) spice mix and leek; (B) spice mix and spice extract; (C) hot pepper, rajan, garlic, chili bean sauce, and leek; (D) hot pepper, spice mix, leek, and spice extract; (E) spice mix and Chinese chive; and (F) spice mix, leek, and spice extract. Because the noodles investigated in this study contain various pungent ingredients, such as hot pepper, rajan, chili bean sauce, garlic, leek, and Chinese chive, the noodle broths are suggested to contain a variety of pungent substances, including capsaicin, dihydrocapsaicin, and nordihydrocapsaicin (present in hot pepper, rajan, and chili bean sauce) and allicin, allyl sulfide, and diallyl sulfide (present in garlic, leek, and Chinese chive). These spicy compounds have been reported as agonists of TRPV1 [5,45,46]. Interestingly, the synergistic effects of TRPV1 agonists on TRPV1 activation have been reported [16,17]. For example, the TRPV1 agonists nicotinic acid, 2-aminoethoxydiphenyl borate, and H^+^ synergistically increase the activation of TRPV1 by capsaicin. These reports suggested that the total TRPV1 activity of a food is not necessarily the sum of the activities of the individual TRPV1 agonists contained therein. Therefore, a method that measures the TRPV1 activity of a whole food, rather than that of separate components, would allow for a more accurate assessment of TRPV1 activity in foods containing multiple pungent compounds.

The present study shows that this luminescence assay system can be used to measure the hTRPV1 activity of spicy foods *per se,* even if they exhibit auto-fluorescence. Furthermore, the hTRPV1 activity assessed by this method correlates strongly with the pungency intensity assessed by human sensory evaluation. Therefore, the luminescence-based hTRPV1 assay system is a powerful tool for the objective quantification of food pungency in the laboratory and industry. Some TRP channels are reported to play important roles in the flavor perception of foods. This assay system might be applicable to the flavor evaluation of foods involving other TRP channels. We successfully applied the luminescence-based system to the hTRPM8 assay to evaluate the coolness of foods (data not shown). Because the luminescence-based assay system developed in this study did not interfere with fluorescence substances contained in foods, this assay system can be applied to various foods, including curry, hot pepper paste, tea, wine, coffee, and juice.

## 4. Conclusions

The aequorin/hTRPV1 assay method developed in this study successfully evaluates the hTRPV1 activity in auto-fluorescent foods, without the need for purification. This luminescence-based assay system has a signal/baseline ratio ten times that of the conventional fluorescence-based method. Furthermore, the hTRPV1 activity level correlates strongly with the pungency intensity measured by sensory analysis. Collectively, these findings indicate that this luminescence-based hTRPV1 assay system will be a powerful tool for objectively, quantitatively, and reproducibly evaluating the pungency of foods in laboratory and commercial settings.

## Figures and Tables

**Figure 1 foods-10-00151-f001:**
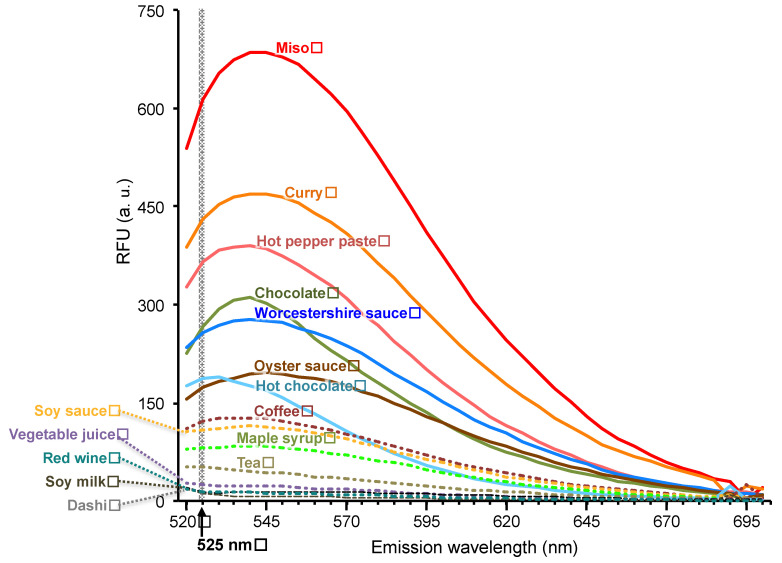
Fluorescence spectra of various foods. Fluorescence spectra upon irradiation with fluorescent light (490 nm). The gray range shows the emission wavelength (525 nm) used for the representative Ca^2+^-sensitive fluorescent probe Fluo-8.

**Figure 2 foods-10-00151-f002:**
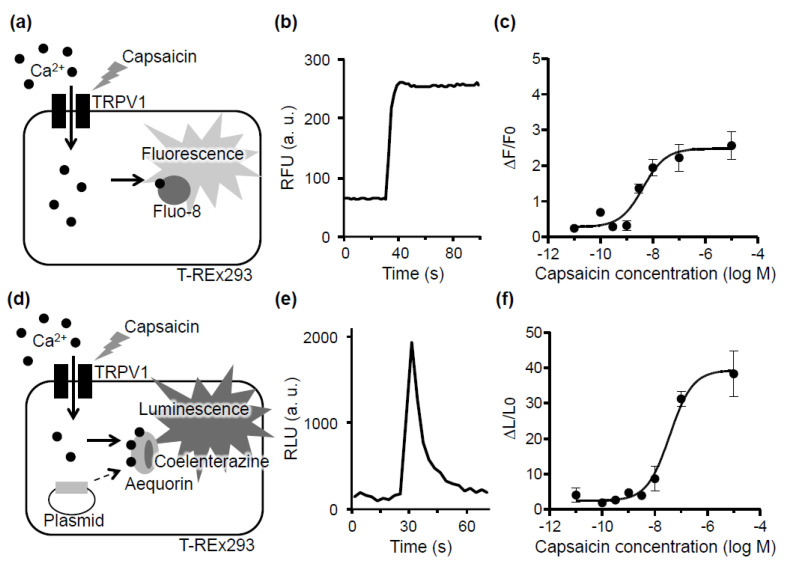
Comparison of measured human transient receptor vanilloid 1 (hTRPV1) activity for fluorescence- and luminescence-based assays. (**a**,**d**) Mechanism underlying the cell-based hTRPV1 assay system for the fluorescence (**a**) and luminescence (**d**) methods. Increased intracellular Ca^2+^ concentration induced by hTRPV1 activation is detected as fluorescent or luminescent signals. (**b**,**e**) Representative Ca^2+^ responses elicited by 10 nM capsaicin in the fluorescence (**b**) and luminescence (**e**) assays. (**c**,**f**) Concentration–response curves for capsaicin on hTRPV1 obtained in the fluorescence (**c**) and luminescence (**f**) methods. Data are presented as the average (*n* = 3) ± SE (error bars) in (**c**,**f**).

**Figure 3 foods-10-00151-f003:**
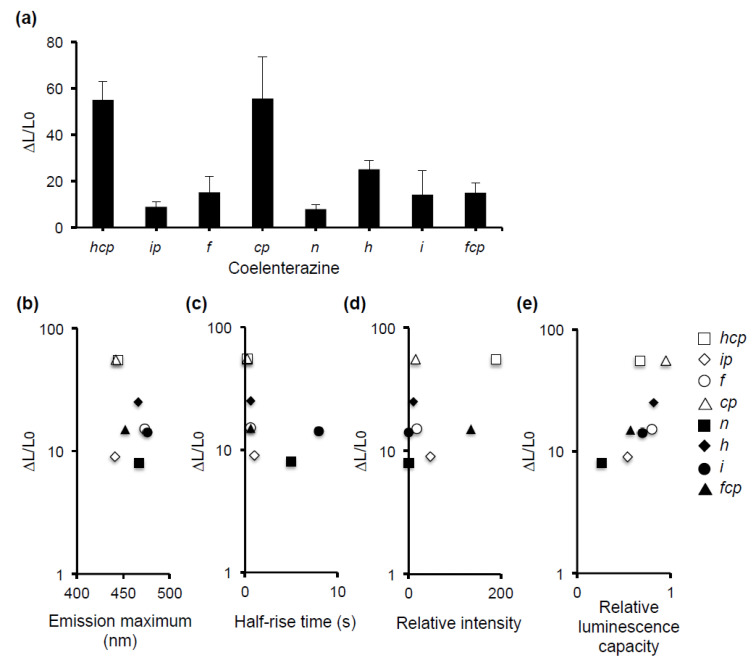
Comparison of the hTRPV1 activity of eight types of coelenterazines with different luminescence properties. (**a**) hTRPV1 activity induced by 10 µM capsaicin for each luminescent substrate. Data are presented as the average (*n* = 3) ± SE (error bars) (**b**–**e**). Correlation between hTRPV1 activity and four luminescence characteristics of coelenterazines. Emission maximum, the maximum emission wavelength; half-rise time, time required for the luminescent signal to reach 50% of the maximum after the addition of 1 mM Ca^2+^; relative intensity, the degree of Ca^2+^ sensitivity relative to that of native aequorin (1.0); and relative luminescence capacity, total time-integrated emission of aequorin in saturating Ca^2+^ relative to that of native aequorin (1.0) [39,40]. Each symbol represents different kinds of coelenterazine (*hcp*, *ip*, *f*, *cp*, *n*, *h*, *i*, and *fcp*). Opened square: *hcp*; opened diamond: *ip*; opened circle: *f*; opened triangle: *cp*; closed square: *n*; closed diamond: *h*; closed circle: *i*; and closed triangle: *fcp*.

**Figure 4 foods-10-00151-f004:**
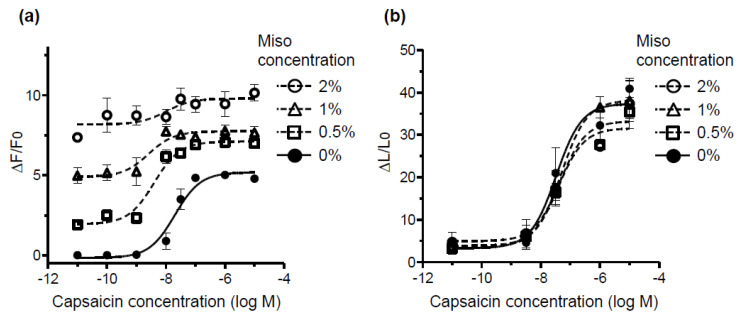
Evaluation of hTRPV1 activity induced by capsaicin using the fluorescence- and luminescence-based assay systems in the presence of miso, which contains fluorescent substances. Dose–response curves for capsaicin on hTRPV1 obtained in the fluorescence (**a**) and luminescence (**b**) assays in the absence and presence of 0.5, 1, and 2% of miso. Data are presented as the average (*n* = 3) ± SE (error bars).

**Figure 5 foods-10-00151-f005:**
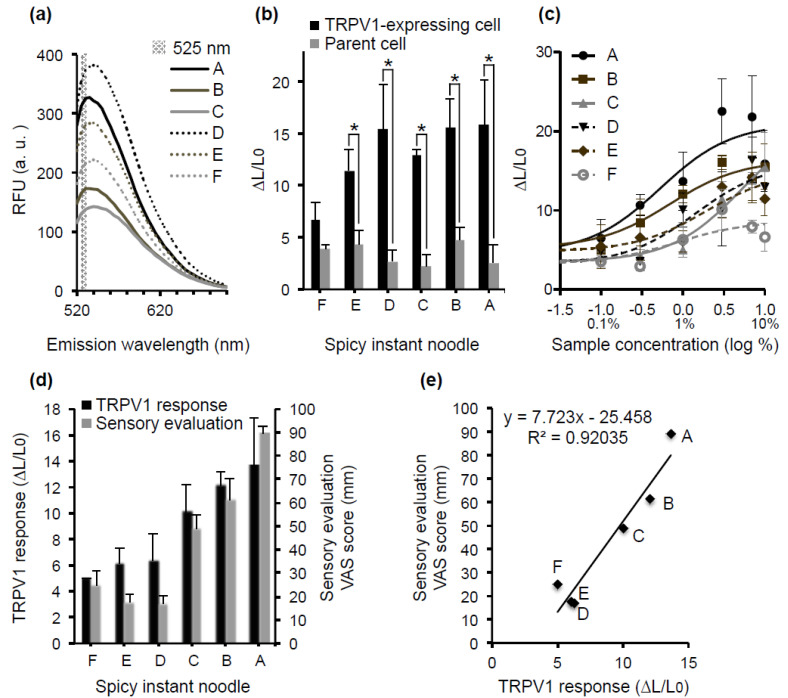
hTRPV1 activity and pungency intensity of spicy instant noodles. (**a**) Six types of spicy instant noodles (designated A–F) were tested. Fluorescence spectra of six kinds of the noodle broths when they were irradiated with fluorescent light (490 nm). The gray range shows the emission wavelength (525 nm) that is used for the fluorescence-based hTRPV1 assay. (**b**) Ca^2+^ responses elicited by 10% noodle broth on the hTRPV1-expressing cells (black) and parent cells not expressing hTRPV1 (gray). Data are presented as the average (*n* = 3) ± SE (error bars); * *p* < 0.05 (unpaired *t*-test). (**c**) Concentration–response curves for hTRPV1 activity induced by noodle broths using the luminescence assay. Data are presented as the average (*n* = 3) ± SE (error bars). (**d**) Correlation between hTRPV1 activity in the luminescence assay (black) and pungency intensity in the sensory test (gray) for 1% and 10% noodle broth, respectively. Data are presented as the average (hTRPV1 assay, *n* = 3; sensory evaluation, *n* = 12) ± SE (error bars). (**e**) Correlation analysis of hTRPV1 activity and the pungency intensity of six noodle broths.

## Data Availability

The data underlying this article are available in the article and in its online Appendix A.

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
