# Peer review of "A Luminescence-Based Human TRPV1 Assay System for Quantifying Pungency in Spicy Foods"

_foods, 2021, doi:10.3390/foods10010151_

Round 1
Reviewer 1 Report
Page 2 of lines 60-64 are more of a conclusion
All abbreviations should be explained when they appear for the first time (even if the authors think that they are commonly known) e.g. page 2 line 68
Fig 1 is hardly legible
The symbols in Fig 3 (b-e) should be clarified / named
Fig 5 b - * Statistically significant differences between what?
Reviewer 2 Report
There are some extremely minor issue with the English grammar. That is really the only thing I have to point out.
Line 26: I would say Pungency, is an oral...
Line 158: instead of using the. You should use their instead.
Reviewer 3 Report
The following items should be revised:
The authors did not provide the aim of the research.
Methods
2.6. Sensory evaluation of spicy instant noodle pungency
It is not clear how many samples the author investigated in each group (the n value is unknown).
Results
Line 291
“with human taste perception.” What taste
Lines 326 - 330
It
Before the manuscript “Luminescence-based human TRPV1 assay system to quantify pungency in spicy foods” acceptation for publication in “Foods” the following items should be revised:
The authors did not provide the aim of the research.
Methods
2.6. Sensory evaluation of spicy instant noodle pungency
It is not clear how many samples the author investigated in each group (the n value is unknown).
Results
Line 291
“with human taste perception.” What taste
Lines 326 - 330
It is not clear whether for all products?
This should be clarified and developed
Comments to the Conclusions:
Lines 338 – 341
“Some TRP channels are reported to play 338important roles in the flavor perception of foods. This assay system might be applicable to flavor 339evaluation in foods involvingother TRP channels. We indeed successfully applied the luminescence-340based system to hTRPM8 assay to evaluate the coolness of foods (data not shown)”
This should be written in results and section of Results and Discussion.
is not clear whether for all products?
This should be clarified
Comments to the Conclusions:
Lines 338 – 341
“Some TRP channels are reported to play 338important roles in the flavor perception of foods. This assay system might be applicable to flavor 339evaluation in foods involvingother TRP channels. We indeed successfully applied the luminescence-340based system to hTRPM8 assay to evaluate the coolness of foods (data not shown)”
This should be written in results and section of Results and Discussion.
Reviewer 4 Report
The paper reads well and but missing minor points
Statistical analysis is missing, please add this section.
Include ethics statement if any for the recruitment of the sensory panel.
There are a total of 12 panellists and it said that 3-4 panellist sits in per booth? Usually, each panellist would receive their own booth, so can the authors explain here what had happened?
How can the authors assure that there is no carry over effect here? It seems that it's just water that is used and we all know that capsaicin is hydrophobic so it wont be rinsed properly.
